# Clinician perspectives on what constitutes good practice in community services for people with complex emotional needs: A qualitative thematic meta-synthesis

Jordan Troup[1], Billie Lever Taylor[2], Luke Sheridan Rains[3]*, Eva Broeckelmann[4], Jessica Russell[4], Tamar Jeynes[4], Chris Cooper[5], Thomas Steare[3], Zainab Dedat[3], Shirley McNicholas[6], Sian Oram[7], Oliver Dale[8], Sonia Johnson[3,6]

1 Health Service and Population Research Department, Institute of Psychiatry, Psychology & Neuroscience, King's College London, London, England, 2 Division of Psychiatry, University College London, London, England, 3 Division of Psychiatry, NIHR Mental Health Policy Research Unit, University College London, London, England, 4 Health Service and Population Research Department, NIHR Mental Health Policy Research Unit Complex Emotional Needs Lived Experience Working Group, Institute of Psychiatry, Psychology & Neuroscience, King's College London, London, England, 5 Department of Clinical, Educational and Health Psychology, University College London, London, England, 6 Camden and Islington NHS Foundation Trust, London, England, 7 Health Service and Population Research Department, NIHR Mental Health Policy Research Unit, Institute of Psychiatry, Psychology & Neuroscience, King's College London, London, England, 8 West London Mental Health Trust, London, England

* l.sheridanrains@ucl.ac.uk

## Abstract

### Introduction

The need to improve the quality of community mental health services for people with Complex Emotional Needs (CEN) (who may have a diagnosis of 'personality disorder') is recognised internationally and has become a renewed policy priority in England. Such improvement requires positive engagement from clinicians across the service system, and their perspectives on achieving good practice need to be understood.

### Aim

To synthesise qualitative evidence on clinician perspectives on what constitutes good practice, and what helps or prevents it being achieved, in community mental health services for people with CEN.

### Methods

Six bibliographic databases were searched for studies published since 2003 and supplementary citation tracking was conducted. Studies that used any recognised qualitative method and reported clinician experiences and perspectives on community-based mental health services for adults with CEN were eligible for this review, including generic and specialist settings. Meta-synthesis was used to generate and synthesise over-arching themes across included studies.

**Data Availability Statement:** The paper reports on a systematic review and therefore draws on papers

that are in the public domain and not on data that we hold. We provide a full list of contributing papers included in this review, all of which can be readily accessed.

**Funding:** This paper presents independent research commissioned and funded by the National Institute for Health Research (NIHR) Policy Research Programme, conducted by the NIHR Policy Research Unit (PRU) in Mental Health (grant no. PR-PRU-0916-22003). The views expressed are those of the authors and not necessarily those of the NIHR, the Department of Health and Social Care or its arm's length bodies, or other government departments. The funders had no role in study design, data collection and analysis, decision to publish, or preparation of the manuscript.

**Competing interests:** The authors have declared that no competing interests exist.

## Results

Twenty-nine papers were eligible for inclusion, most with samples given a 'personality disorder' diagnosis. Six over-arching themes were identified: 1. The use and misuse of diagnosis; 2. The patient journey into services: nowhere to go; 3. Therapeutic relationships: connection and distance; 4. The nature of treatment: not doing too much or too little; 5. Managing safety issues and crises: being measured and proactive; 6. Clinician and wider service needs: whose needs are they anyway? The overall quality of the evidence was moderate.

## Discussion

Through summarising the literature on clinician perspectives on good practice for people with CEN, over-arching priorities were identified on which there appears to be substantial consensus. In their focus on needs such as for a long-term perspective on treatment journeys, high quality and consistent therapeutic relationships, and a balanced approach to safety, clinician priorities are mainly congruent with those found in studies on service user views. They also identify clinician needs that should be met for good care to be provided, including for supervision, joint working and organisational support.

## Introduction

The global prevalence of "personality disorder" in the community is estimated to be around 7.8% [1]. This increases to between 40 and 92% among people who use community secondary mental health care services in Europe [2]. High rates of comorbidity with other mental health conditions have been identified [3,4] and people with comorbid conditions appear to have particularly high inpatient and involuntary service use and poor outcomes [5,6]. High rates of comorbid physical conditions have also been found [7–9] and evidence suggests shorter life expectancies [10]. Impacts on quality of life are comparable to serious somatic illness [11] and a substantial economic cost has been found for health and social care services and society more generally [12,13].

Our team, which includes people with relevant lived experience and clinicians, has debated terminology for this review in light of rapidly evolving debates about the term 'personality disorder' (especially 'borderline personality disorder'). While some service users report that they find it helpful in clarifying the nature of their difficulties and it has a role in ensuring consistency in research, very serious critiques have been made of this diagnosis as stigmatising, potentially misogynistic, and associated with a lack of hope and of progress in delivering effective care [14–18]. Many service users find it unhelpful and do not identify with it. For this reason, in this paper and our companion papers on this topic, we have chosen to use the term complex emotional needs (CEN) as a working description of the cluster of needs that may lead to a "personality disorder" diagnosis, and / or to using services for 'personality disorder' or CEN, or who appear to have similar needs (e.g., related to repeated self-harm). It is not our intention that complex emotional needs becomes a substitute diagnosis, but rather a description of a broad group of service users. We advocate co-produced work to develop new ways of describing and assessing their difficulties that are clear, consistent and acceptable. While we use the term CEN in our summary of themes from the papers, as the tables of supporting materials indicate, most of the papers themselves use the term "personality disorder".

In the UK, care provided for people with CEN has recurrently been described as of very variable and often poor quality [19]. In 2003, new policy guidance was published aimed at greatly increasing provision of specialist services and improving training and support in generic services [20,21]. The number of mental health Trusts providing dedicated services increased five-fold over the following decade, but a national survey in 2015 found persisting deficits in access to specialist therapies and to a full spectrum of biopsychosocial interventions, and it remained unclear as to whether overall quality of care had improved [19]. Improving care for people with CEN has since become a renewed priority in England [22–24]. The need to improve quality of care, reduce stigma and deliver effective treatments for CEN is recognised internationally [25], with formulation of policies and guidelines in various countries aimed at improving care [26,27].

Policy focus on improving CEN care has been accompanied by growing evidence that there are effective psychological treatment options for CEN [28–33], but that the translation of policy and evidence into service provision has been slow [34]. Service users and clinicians have been found to agree that access to specialist services and psychological interventions, interventions to reduce stigma in services, specialist consultation services for generic mental health staff, and positive risk management are priorities, but these do not appear to be widely reflected in service provision [35].

As well as lack of resources, clinician-related barriers to service improvement have repeatedly been found. Stigma related to "personality disorder" diagnosis has recurrently been identified among clinicians: feeling powerless to be helpful, perceived un-treatability, preconceptions about patients and poor CEN understanding have been identified as contributors to this stigma [34,36]. Unmet training needs and lack of a clear framework are reported to contribute to negative experiences of working with people with CEN [37]. These do not, however, appear to be inevitable consequences of working with CEN: in a relatively well-resourced specialist "personality disorder" service setting, Crawford et al. [38] reported relatively low levels of clinician burnout and good satisfaction among staff working with people with CEN. Thus, understanding the perspectives, experiences and attitudes of clinicians and the conditions that allow them to work effectively and without excessive burnout with people with CEN is a crucial element in informing next steps for improving service provision.

The aim of this review was to synthesise existing qualitative evidence on clinician perspectives on what constitutes good practice in community mental health settings for people with CEN, and how this could be achieved. Objectives included conducting a systematic search of the literature, conducting a meta-synthesis of qualitative data, and assessing the quality of the evidence. This review is part of a broader programme of work conducted by the NIHR Mental Health Policy Research Unit to inform the development of NHS England specialist pathways and to strengthen the evidence base for service development in this field nationally and internationally. Other reviews include a synthesis of qualitive literature on service user perspectives on good practice [17], systematic reviews on treatment effectiveness and cost-effectiveness [39] and a study of service typologies.

## Methods

### Information sources and search strategy

The review team developed the protocol in line with PRISMA guidelines [40] and guidance on qualitative meta-syntheses [41] in collaboration with a project-specific working group of lived-experience researchers and subject experts. The protocol was registered prospectively on PROSPERO (CRD42019145615), as was the protocol for the wider programme of work (CRD42019131834).

One search strategy was developed for all the reviews in the programme (see S1 Appendix). Search terms were built around key words and subject headings relevant to CEN and related needs, community mental health services, and eligible study designs including qualitative, quantitative and guidelines. Comprehensive searches were conducted of MEDLINE (January 2003—December 2019), Embase (January 2003—December 2019), HMIC (January 2003—December 2019), Social Policy and Practice (January 2003—December 2019), CINAHL (January 2003—December 2019) and ASSIA (January 2003—January 2019). No limits were placed on the language or country, and a limit of 2003 or later was placed on the date to capture perspectives of greater contemporary relevance by only including research since the release of "Personality Disorder: No Longer a Diagnosis of Exclusion" and National Institute of Mental Health in England policy implementation guidance [20,21].

Citations retrieved during searches were collated in Endnote [42], a reference management software, and duplicates were removed. Titles and abstracts were double screened by two NIHR Mental Health Policy Research Unit researchers for all the reviews together and full text screening was performed on potentially eligible papers for this review. Supplementary searching included a call for evidence publicised via the study team's networks, relevant professional associations and social media, forward and backward citation tracing of included articles, and reference lists of other relevant systematic reviews found in an additional systematic review search of EMBASE and MEDLINE (January 2003—November 2019). Grey literature was identified through web searches and the above bibliographic database search. All included studies and 20% of those excluded were double screened, and discussion with senior reviewers achieved consensus.

## Eligibility criteria

Studies using recognised qualitative data collection and analysis methods to explore clinician perspectives on good practice in community mental health services for people with CEN were included. For the purposes of this paper, we have defined good practice as that which is likely to contribute to or be associated with improved service user outcomes, experiences and satisfaction with services. Studies were eligible if they reported the relevant perspectives of any mental health professional with experience of working with people with CEN. Our main sample was of publications which used the term 'personality disorder' to describe the difficulties discussed by clinicians. However, we were aware that other investigators may also have wished to avoid the term 'personality disorder' or may have collected data from people who did not identify with this diagnosis but were experiencing comparable long-term difficulties. As more fully detailed in S1 Appendix, we also ran searches with other terms which might be used to describe such difficulties (for example, recurrent self-harm, complex trauma, emotion dysregulation). As described below, this however yielded few papers. Eligible settings were community-based mental health services, i.e. any non-residential mental health services that provided care for people living in the community with CEN, whether exclusively or not. This included mental health care in primary care settings, generic community mental health teams (e.g., mainstream multidisciplinary teams providing services for a range of needs in the local population), and specialist/dedicated services exclusively for people with CEN [19]. Residential, forensic, or crisis services, or specialist services for different conditions were excluded. Papers were excluded if the service target population were primarily below the age of 16, unless focussing on transition into adult services. Initially, peer-reviewed and grey literature were eligible, except for case studies, dissertations and theses. Due to the broad scope in topics covered, a pragmatic decision was made ad-hoc to exclude papers not in English and not peer-reviewed (See S2 Appendix for full eligibility criteria). Most of the papers used "personality disorder" to

describe the sample, but here we use the term CEN as an overall term for reasons discussed in the introduction.

## Quality assessment and analysis

Study characteristics were extracted into a Microsoft Excel form. The Critical Appraisal Skills Programme (CASP) Qualitative Checklist [43] was used to perform quality assessments. Study quality was not used to determine eligibility but is reported below. Text from results sections of included articles was entered verbatim into the coding software NVivo for thematic meta-synthesis [41] and linked to individual study characteristics such as types of clinicians, services, and interventions. For stage one, articles were coded line-by-line by one of two researchers and 20% of papers were double coded to produce an initial framework. A preliminary thematic framework emerged from further discussion between the two researchers for stage two, developed as codes were merged and grouped hierarchically. At stage three, analytic themes were developed and finalised iteratively through wider collaboration with the team of reviewers and experts by experience and occupation. The analysis process included considering whether there were sub-group differences related to major study characteristics such as country of publication.

## Results

A total of 29 papers (drawing on 27 unique datasets) were eligible for inclusion [38,44–71] (Fig 1), representing perspectives from at least 550 clinicians. Clinicians represented a variety of professions, including (but not limited to) psychologists (8 papers), social workers (7), psychiatric nurses (12), occupational therapists (4), psychiatrists (12), family doctors (known as General Practitioners or 'GPs' in the UK; 3) and counsellors (2). Other papers defined their clinicians more broadly as those who provide the service of interest and some samples also included service managers, commissioners, administrators and referrers. Twelve studies were conducted in generic community mental health settings, four in primary care, seven in specialist / dedicated services for people with CEN and three in specific DBT-teams, with a further three studies including clinicians from across a range of community settings. The majority of included papers came from England (13), followed by Australia (5) and North America (3). Data collection methods were sometimes mixed and consisted of primarily interviews (22), focus groups (7), and open-text responses / surveys (4). While we use the term "CEN" in our summary, service users in most included studies were identified as having "personality disorder" or "borderline personality disorder", as summarised in Table 1 below.

Quality appraisal indicated that the majority of studies appropriately used qualitative methodology (n = 28), employed an appropriate research design (n = 28), and described clear findings (n = 28). Most studies also presented clear aims (n = 27) and used appropriate data collection methods (n = 26). However, a number of papers did not provide enough information to determine whether the data analysis was sufficiently rigorous (n = 6), whether the recruitment strategy was appropriate (n = 11), nor whether ethical issues had been sufficiently considered (n = 12). Only 5 papers in total adequately considered the relationship between researcher and participants. See S1 Table for full appraisal ratings.

Six overarching themes were identified through meta-synthesis: 1. Stigma and the use and misuse of diagnosis; 2. The patient journey through services: nowhere to go; 3. Therapeutic relationships: connection and distance; 4. Dialectics: not doing too much or too little; 5. Managing safety issues and crises: being measured and proactive; and 6. Clinician and wider service needs (including clinician support, interagency working and the wider system, and establishing new services, interventions and skills): whose needs are they anyway? These themes are

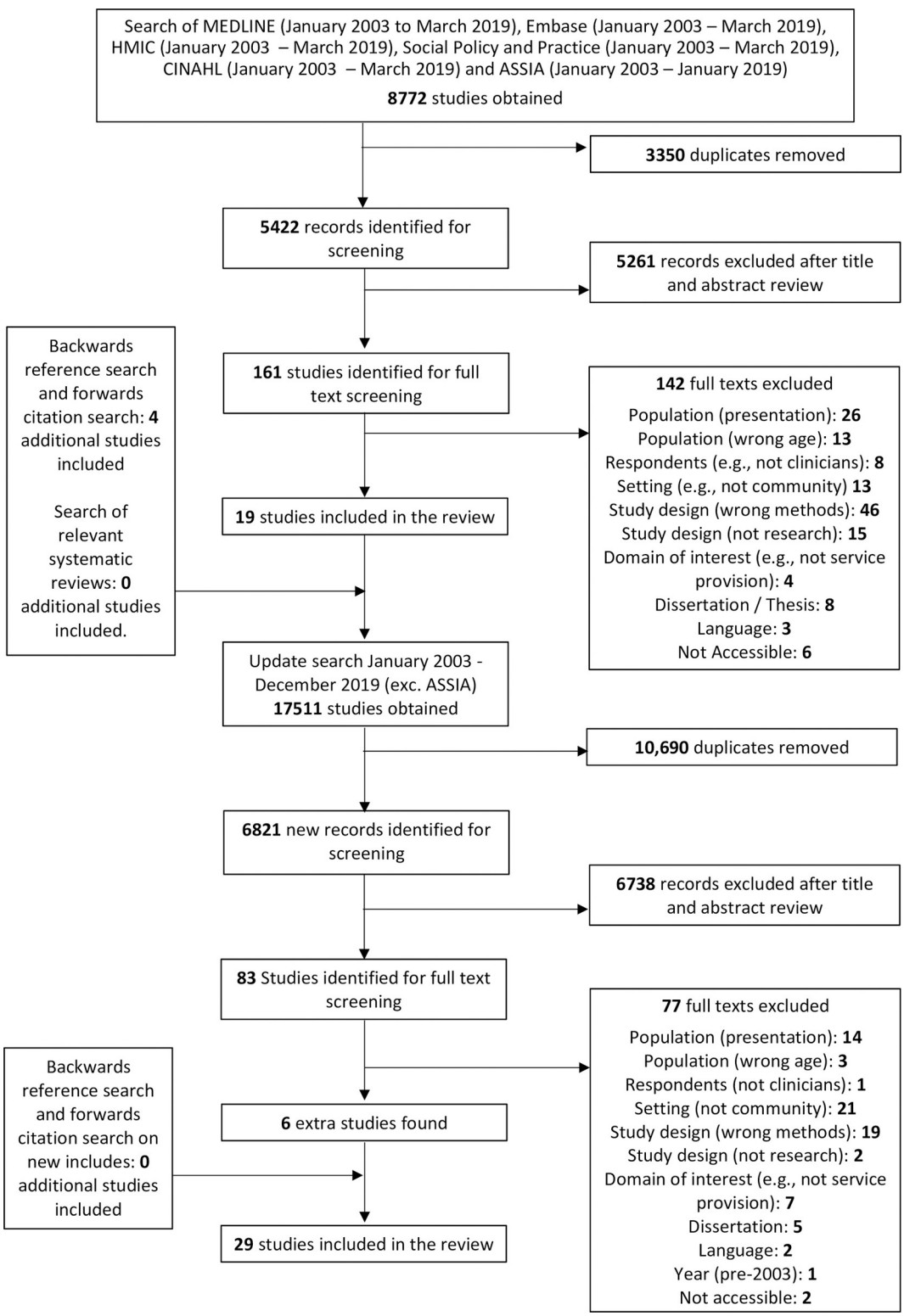

**Fig 1. PRISMA Diagram.**

**Table 1. Study characteristics.**

*First Author, Year. Title.*

| Clinician type Sample size | Data collection | Service / setting, Location | Target population | Intervention if applicable |
|---|---|---|---|---|
| *Bosanac, 2015. [44] Mentalization-based intervention to recurrent acute presentations and self-harm in a community mental health service setting.* | | | | |
| Case managers: psychiatric nurses and occupational therapists N = 8 | Five 3-mothly focus groups of 3–5 clinicians | Community mental health service, Australia | 8 female service users diagnosed with 'BPD' (DSM-IV) and <7 on DIB-R. | MBI |
| *Carmel, 2014. [45] Barriers and solutions to implementing dialectical behavior therapy in a public behavioral health system.* | | | | |
| Clinicians NS N = 19/34 | Structured phone interviews | Community mental health and substance abuse agencies within a public behavioral health system, Northern California | People with 'BPD' | DBT |
| *Crawford, 2007. [47] Learning the lessons: a multi-method evaluation of dedicated community-based services for people with personality disorder.* | | | | |
| Service managers, front-line clinicians (range), referrers, commissioners. N = 89 service providers, 26 referrers, 13 commissioners from across all 11 dedicated services. | Comprehensive evaluation including in-depth qualitative interviews | 11 'Pilot' dedicated services, England | People with 'PD'–range of criteria across services | Range– psychotherapeutic, social, occupational |
| *Crawford, 2007. [46] Lessons learned from an evaluation of dedicated community-based services for people with personality disorder.* | | | | |
| See Crawford 2007a, above | | | | |
| *Crawford, 2010. [38] Job satisfaction and burnout among staff working in community-based personality disorder services.* | | | | |
| Service managers and 'front-line' clinicians: therapists, psychotherapists, nurses, psychologists, social workers, psychiatrists, occupational therapists, art therapists, support workers, and employed service users. N = 89 from across all 11 dedicated services | Comprehensive evaluation including in-depth qualitative interviews | 11 'Pilot' dedicated services, England | People with 'PD' | Range: psychotherapeutic, social, occupational |
| *Donald, 2017. [48] Clinician perspectives on recovery and borderline personality disorder.* | | | | |
| Social workers, nurses, psychologists, one psychiatry registrar and one consultant psychiatrist N = 16 | Interviews | Clinicians mostly from one specialist service, and two from a generalist service, Australia | People with 'PD' / 'BPD' | Range |
| *Fanaian, 2013. [49] Improving services for people with personality disorders: Views of experienced clinicians.* | | | | |
| Recognised specialists and experts in 'PD' N = 60 | Written group responses to one question during clinical and scientific meeting | Range of public and private services across Australia | NA | NA |
| *French, 2019. [50] GPs' views and experiences of managing patients with personality disorder: a qualitative interview study.* | | | | |
| General practitioners N = 15 | Phone interviews with topic schedule | GP Practices, West of England | People suspected by GP to have 'PD' | NA |
| *Herschell, 2009. [51] Understanding community mental health administrators' perspectives on dialectical behavior therapy implementation.* | | | | |
| Mental health service administrators N = 13 from 9/10 participating organisations | Semi-structured phone interviews | Ten provider organisations partnered with a large non-profit managed behavioral health organization, Pennsylvania | Primarily people with 'BPD', some other disorders | DBT |
| *Hogard, 2010. [52] An evaluation of a managed clinical network for personality disorder: breaking new ground or top dressing?* | | | | |
| Network staff from across multiple agencies with diverse backgrounds, including psychotherapy, occupational therapy, and advocacy N = All staff from network | Semi-structured interviews | A managed clinical network for 'PD', England | People with a diagnosis of 'PD' | NA |
| *Hutton, 2017. [53] Switching roles: a qualitative study of staff experiences of being dialectical behaviour therapists within the National Health Service in England.* | | | | |

*(Continued)*

**Table 1.** (*Continued*)

*First Author, Year. Title.*

| Clinician type Sample size | Data collection | Service / setting, Location | Target population | Intervention if applicable |
|---|---|---|---|---|
| Clinicians from 3 DBT teams: social workers, community psychiatric nurses and clinical psychologists N = 6/24 from all 3 teams | Semi-structured interviews | 3 DBT teams within 1 Trust (alongside secondary care service roles), England | People with difficulties associated with 'BPD' | DBT |

*Koekkoek, 2009. [54] Clinical problems in community mental health care for patients with severe borderline personality disorder.*

| Expert mental health professionals from different disciplines, different treatment locations, and different educational backgrounds, with expertise on treatment for people with 'BPD' and at least 3 years' experience N = 8 | Focus group | Experts had at least some experience with the specialised treatment of such patients, but worked in a general setting | Severe 'BPD' (DSM-IV) | NA |

*Lamph, 2019. [55] Personality disorder co-morbidity in primary care 'Improving Access to Psychological Therapy' services: A qualitative study exploring professionals' perspectives of working with this patient group.*

| Trained and trainee psychological wellbeing practitioners, high intensity cognitive behavioural therapist, clinical psychologists, clinical leaders and IAPT clinical service managers N = 28 | Interviews | IAPT in 2 localities (primary care), England | People with CMD and co-morbid 'PD' | IAPT interventions e.g., CBT |

*Langley, 2005. [56] Trust as a foundation for the therapeutic intervention for patients with borderline personality disorder.*

| Multidisciplinary clinicians with extensive experience in the management of 'BPD' in both private and public systems: psychiatrists, psychiatric nurses, a psychiatric social worker, a clinical psychologist and a counselling psychologist N = 10 | Individual interviews or focus group | Psychiatric Community Services, South Africa | People with 'PD' (DSM-IV) | NA |

*Lee, 2008. [57] A pilot personality disorder outreach service: development, findings and lessons learnt.*

| Consultant psychiatrists N = Unclear. 13 SUs in case series element, unknown number of psychiatrists from across 8 teams, from which 2 were selected for outreach service. | Semi-structured interviews | Pilot 'PD' outreach service (secondary care), England | People with 'BPD' (SAP) | MBI / psychodynamic |

*Morant, 2003. [58] A multi-perspective evaluation of a specialist outpatient service for people with personality disorders.*

| Referrers to service: consultant psychiatrists, social workers, one clinical psychologist, one substance misuse worker, and one clinical nurse specialist N = 12 | Multi-perspective / multi-method evaluation including semi-structured interviews | Specialist 'PD' outreach service (clinicians primarily from CMHTs), England | People with moderate to severe 'PD' | Individual treatment (cognitive therapy), Group psychotherapy (psychodynamic), Art psychotherapy (group) |

*O'Connell, 2013. [59] Community psychiatric nurses' experiences of caring for clients with borderline personality disorder.*

| Psychiatric nurses N = 10 | Interviews | CMHT (secondary care), Ireland | People with 'BPD' | |

*Perseius, 2003. [61] Treatment of suicidal and deliberate self-harming patients with borderline personality disorder using dialectical behavioral therapy: the patients' and the therapists' perceptions.*

| DBT therapists: a psychiatrist, a registered nurse, and cognitive psychotherapists N = 4/4 | Individual free-format questionnaire and group interview | DBT Team, Sweden | People with 'BPD' or related symptoms | DBT |

*Perseius, 2007. [60] Stress and burnout in psychiatric professionals when starting to use dialectical behavioural therapy in the work with young self-harming women showing borderline personality symptoms.*

(*Continued*)

**Table 1.** (Continued)

*First Author, Year. Title.*

| Clinician type<br>Sample size | Data collection | Service / setting, Location | Target population | Intervention if applicable |
|---|---|---|---|---|
| Physicians, psychologists, registered nurses, mental health care assistants and one occupational therapist<br>N = 22 | An individual open question, free text answer questionnaire and a group interview (and burnout inventory) | DBT Team, Sweden | Women with 'BPD' | DBT |

*Pigot, 2019. [62] Barriers and facilitators to the implementation of a stepped care intervention for personality disorder in mental health services.*

| | | | | |
|---|---|---|---|---|
| Mental health clinicians and managers actively involved in the intervention<br>N = 21/46 | Semi-structured interview | Publicly funded open access provider of health and medical services, Australia | People with 'PD', particularly 'BPD' | Stepped-care approach |

*Priest, 2011. [63] How can mental health professionals best be supported in working with people who experience significant distress?*

| | | | | |
|---|---|---|---|---|
| Social workers, nurses, occupational therapists, psychiatrists, psychologists and support workers<br>N = 26 | Focus groups | CMHTs and CSMT (secondary care), England | CMHT case load / People with 'PD' / people who experience 'significant distress' | NA |

*Rizq, 2012. [64] 'There's always this sense of failure': an interpretative phenomenological analysis of primary care counsellors' experiences of working with the borderline client.*

| | | | | |
|---|---|---|---|---|
| Experienced counsellors (senior practitioners)<br>N = 5 | Semi-structured interviews | Primary care, England | People with 'BPD' (clinician judgement) | NA |

*Stalker, 2005. [65] It is a horrible term for someone': service user and provider perspectives on 'personality disorder'.*

| | | | | |
|---|---|---|---|---|
| Psychiatrists, three community psychiatric nurses, one clinical psychologist, one senior social worker and one senior occupational therapist, managers and an administrator<br>N = 12 | Interviews | CMHTs through Mental Health Resource Centres, Scotland | People with 'PD' | NA |

*Stroud, 2013. [66] Working with borderline personality disorder: A small-scale qualitative investigation into community psychiatric nurses' constructs of borderline personality disorder.*

| | | | | |
|---|---|---|---|---|
| Community psychiatric nurses<br>N = 4 | Semi-structured interviews | CMHT, Wales | People with 'BPD' | NA |

*Sulzer, 2016. [67] Improving patient-centered communication of the borderline personality disorder diagnosis.*

| | | | | |
|---|---|---|---|---|
| Psychiatrists, Psychologists, Clinical Social Workers and BPD activists<br>N = 32 | Semi-structured interviews | Clinicians from 11 states, America | People with 'BPD' | NA |

*Thompson, 2008. [68] Multidisciplinary community mental health team staff's experience of a 'skills level' training course in cognitive analytic therapy.*

| | | | | |
|---|---|---|---|---|
| All eligible clinicians: social workers and community psychiatric nurses<br>N = 12 | Structured, open-ended interviews | CMHT (secondary care), UK | People with complex needs e.g., people presenting with features of 'PD' | CAT |

*Vyas, 2017. [69] Working in a therapeutic community: exploring the impact on staff. Therapeutic Communities: The International Journal of Therapeutic Communities.*

| | | | | |
|---|---|---|---|---|
| Clinicians working in a TC<br>N = 8 | Semi-structured interviews | A long-standing TC, UK | People with 'EUPD' / 'emotional instability' | CAT / MBT |

*Wilson, 2018. [70] Experiences of parenting and clinical intervention for mothers affected by personality disorder: a pilot qualitative study combining parent and clinician perspectives.*

| | | | | |
|---|---|---|---|---|
| Referring CAMHS clinicians<br>N = 5 | Semi-structured interviews | Four CAMHS teams referred into the Helping Families Programme, England | Mothers with 'PD' who had a child (living with them) aged 3–11 years with a behavioural and/or emotional disorder | Helping Families Programme–parenting and clinical intervention |

*Wlodarczyk, 2018. [71] Exploring General Practitioners' Views and Experiences of Providing Care to People with Borderline Personality Disorder in Primary Care: A Qualitative Study in Australia.*

(*Continued*)

**Table 1.** (Continued)

*First Author, Year. Title.*

| Clinician type Sample size | Data collection | Service / setting, Location | Target population | Intervention if applicable |
|---|---|---|---|---|
| Any currently practicing GPs N = 12 | Focus groups | Primary care, Australia | People with 'BPD' | NA |

Abbreviations: NS = Not Specified. BPD = Borderline Personality Disorder. DSM-IV = Diagnostic and Statistical Manual of Mental Health Disorders Version 4. DIB-R = Diagnostic Interview for Borderline Patients–Revised. MBI/MBT–Mentalisation Based Intervention / Therapy. DBT = Dialectical Behavioural Therapy. PD = Personality Disorder. GP = General Practitioner. IAPT = Improving Access to Psychological Therapies. CMD = Common Mental Disorders. CBT = Cognitive Behavioural Therapy. SU = Service User. SAP = Standardised Assessment of Personality. CMHT = Community Mental Health Team. CSMT = Community Substance Misuse Team. CAT = Cognitive Analytic Therapy. TC = Therapeutic Community. EUPD = Emotionally Unstable Personality Disorder. CAMHS = Child and Adolescent Mental Health Service.

further described below. S2 Table gives further supporting quotes from the studies relevant to each theme. While conducting the analysis, variations by setting or by participant characteristics were considered. Substantial variations by country or by year of data collection were not identified but variations between types of clinician and service setting were found: these are described where relevant.

## Stigma and the use and misuse of diagnosis

Our main aim was to synthesise evidence on clinician views of good practice, but it was clear that underlying beliefs about the nature of such difficulties and the appropriate use of diagnosis influenced clinicians' perspectives on care. A few studies reported that some clinicians found conceptualising and diagnosing difficulties as "personality disorder" helpful. They saw it as offering a 'common language', and a useful way to understand service users' difficulties, while also helping to ensure that service users were seen as having genuine needs.

However, across a number of studies, clinicians questioned the use, meaning and validity of this diagnosis. They saw it as being associated with stigma, discrimination and exclusion from services, felt it could be difficult to 'shake off', and risked becoming "the person's entirety" [48].

*Patients with a psychosis were seen as not accountable and in need of support. Borderline patients, however, were considered theatrical, posing, and in need of punishment.*

Psychologist describing a crisis intervention team (Koekkoek et al., 2009) [54]

Accounts of the use of "personality disorder" diagnoses in non-specialist primary and secondary care services suggested it was made at times on a basis of "gut instinct" [55] or "gut feeling" [71] or because other diagnoses did not 'fit'. An investigation of clinician views in generic community and voluntary sector services found that some perceived "personality disorder" as essentially "a form of social deviance or cultural rule-breaking" [65], while others felt that the label was an unhelpful medicalisation of legitimate feelings of distress, especially among women. In this study, as in several studies examining perspectives of specialist clinicians, a majority of clinicians saw trauma and adversity as major causes of "personality disorder". As a result of concerns about diagnosis, clinicians were reported in several studies to be reluctant to use this label and to avoid discussing it with service users. Some opted for alternative diagnoses (e.g., complex post-traumatic stress disorder) or employed what they considered to be 'euphemisms' like "difficulty managing emotions" [67]. Other specialist clinicians reported that they preferred a focus on narrative descriptions of presenting difficulties rather than relying on a "personality disorder" diagnosis.

## 1. The patient journey into services: nowhere to go

Access to services for people with CEN was reported in several studies to be a persistent difficulty, with GPs in one study [50] reporting longer waiting times than for any other group of mental health service users. Referrals for specialist support were impeded by factors such as a lack of local services, lack of awareness of services, frequent changes to services, and poorly established referral pathways. This was felt to risk disengagement, escalation of distress, or missing windows of opportunity to provide effective support.

Thresholds for acceptance by specialist services were reported in some studies to be inconsistent and influenced by subjective judgements regarding for example 'severity', 'stuck-ness' or 'motivation to engage'. Many service users were excluded from specialist support due to being perceived as a risk to others (e.g., through having a forensic history), having substance misuse problems, exhibiting behaviour considered too 'problematic' or 'chaotic', or being seen as 'non-psychologically minded'.

Referrers such as GPs in several studies also reported difficulties getting service users accepted by generic, mainstream community mental health teams or psychological treatment services. However, in other studies, clinicians working in these generic teams saw their eligibility criteria as over-inclusive, with one study describing them as a "dumping ground" for anyone who did not 'fit' elsewhere [63]. Stepped care pathways could also contribute to difficulties accessing appropriate treatment. For example, clinicians in the UK reported being encouraged to refer initially to primary care Improving Access to Psychological Therapy (IAPT) or mainstream secondary care services, rather than to specialist teams. However, knowledge and capacity for treating CEN were often seen as lacking in these generic services, with people with CEN not prioritised and clinicians feeling they did not have the skills to deliver expected care. Some referrers described 'embellishing' referral information to meet thresholds for specialist support. However, in other cases, GPs as well as assessors in secondary care services, 'downplayed' service users' difficulties or risk levels and emphasised 'more agreeable' traits to meet thresholds for primary care support, such as IAPT services. Service users could end up being passed back and forth in "a tennis ball effect" [55] with a high but inefficient use of services.

> *You know if you mention 'PD' there will be nowhere at all for them to go so I'm usually very careful not to put it down in their notes. I usually say depressed or a bit anxious. Something that won't make them think the patient is risky. It's about knowing the hoops that you've got to jump through.*

GP (French et al., 2019) [50]

The referral process was reported to be facilitated by good working relationships and communication between receiving clinicians and referrers, outreach by specialist services to raise visibility and explain service models, and acceptance of self-referrals, which some felt could be empowering and inclusive. Some referrers valued holistic, in-depth assessments and formulations from specialist clinicians, particularly non-medical, non-psychiatric or psychodynamic formulations, even if service users ultimately weren't taken on, as these could inform treatment plans and facilitate therapeutic relationships.

## 2. Therapeutic relationships: Connection and distance

Strong, trusting relationships between clinicians and service users were seen as key to treatment success across many studies, but clinicians' experiences of such relationships varied greatly both between and within studies. In several studies, clinicians were keen to emphasise the positives of working with people with CEN, describing them as 'relatable', 'honest' and

'creative', and seeing the role of the clinician as being to "harness that" [38]. However, negative feelings and a sense of burnout were also frequently described, with clinicians viewing (or reporting that other clinicians viewed) service users as 'demanding', 'challenging', 'risky', 'dependant', 'self-destructive', 'manipulative', 'non-compliant', 'untreatable', and likely to 'push boundaries'. Service users' difficulties were seen as enduring but urgent, and clinicians could feel overwhelmed by "a bottomless pool of need" [71] especially as comorbid diagnoses and wider social issues with housing, employment, finances and social networks were often also present. Clinicians described feeling both idealised by service users and as though nothing they did was good enough. While establishing an authentic connection with service users was seen as vital, clinicians admitted to fears of being "sucked dry" and "emotionally swamped" [64], experiencing feelings of vulnerability and of being dangerously on the edge of losing their sense of self.

> *Participants spoke repeatedly about the need to maintain a psychological distance from clients in order to prevent themselves from becoming overwhelmed or burned out.*
>
> (Langley & Klopper, 2005) [56]

In a few studies, however, clinicians reported they felt able to make use of their unsettling feelings to connect with service users' own feelings. Although there were exceptions, negative attitudes and experiences appeared particularly prevalent in mainstream primary and secondary care services. This was attributed to poor understanding of CEN in these settings, to staff being overburdened but inadequately supported, and to observing poor outcomes, leading to frustration, hopelessness, and sometimes feelings of aggression and blame towards service users. Suggestions to combat negative attitudes included better supervision and training by specialists to improve understanding, compassion, and perceptions of treatment effectiveness, along with more support from services for clinicians to engage with supervision and training.

Overall, the impression across studies was that clinicians described the need to be authentic, non-judgemental, empathic, collaborative, hopeful, motivating, consistent and dependable to build trust with service users, whom they understood often to have had histories of abuse or abandonment by key attachment figures. The importance of 'knowing' service users, holding them in mind, and acknowledging the reality of their experiences was emphasised. When relationships went well, clinicians described successfully negotiating connection and distance in the therapeutic relationship: being open, warm and available, but also retaining boundaries, structure and a degree of emotional detachment. Clinicians spoke of a need to create a sense of shared responsibility for progress with service users, and of the value of adopting a curious, non-expert stance to help develop a safe space where strong emotions could be processed, tolerated and "radically accepted" [48].

Clinicians who reported more positive relationships tended to be those who felt better supported, for example describing better team working, supervision, and informal support from their colleagues, as well as longer-term treatment frameworks, which allowed time for relationships to develop. Such support appeared to be much more available in more specialised services.

### 3. Dialectics: Not doing too much or too little

Clinicians' beliefs regarding appropriate duration of treatment, and how best to negotiate not doing 'too much' or 'too little', were complex. There was consensus across studies that people with CEN had long-term needs, but in a few studies, clinicians voiced concerns that open-ended, long-term support could be too demanding for service users to engage with, too resource-intensive, or could result in 'dependency' and a lack of delivery of interventions with

clear therapeutic content, particularly in generic secondary care services. Clinicians felt that it was important to be realistic about what they could achieve and to avoid setting expectations that they could 'fix' everything. At the same time, in several studies, clinicians emphasised that not offering sufficient long-term support could result in unrealistic expectations for recovery, disappointment and undertreatment. Several studies reported a perceived lack of well-developed, longer-term support programmes at a medium level of intensity.

> *The requirements of the system do not always fit with the needs of the people who are using the service: The expectation is that you will recover. . . you will get out of the service. . . we can only work with you for a certain amount of time. . . It just doesn't work as simply as that.*

> Mainstream secondary care clinician (Priest et al., 2011) [63]

Across studies, clinicians described a need for balance between recognising the limits of what could be achieved, managing the expectations of both clinicians and service users, and maintaining hope. In one study, clinicians saw a tendency in mainstream settings for clinicians to "do completely nothing" [54] in therapeutic encounters with people with CEN, or alternatively to display 'false optimism' or 'therapeutic nihilism', rapidly discharging service users due to underlying feelings of powerless and demoralisation. Paradoxically, however, such undertreatment then had the effect of increasing the very 'dependency' clinicians feared, as service users had to keep 'coming back for more'.

Premature discharge was identified as common and was put down to clinicians seeking to 'escape' from work they found challenging, to service recovery models conflicting with service users' needs, and to pressures to move people on. Yet, there was consensus across several studies that discharge could be particularly challenging for people with CEN and needed to be managed sensitively, especially because of associated safety issues (e.g., due to service users feeling abandoned by clinicians). Views diverged, however, about the best way to approach discharge. For example, in one study evaluating specialist services for CEN [47], some clinicians feared that open-ended service use without a clear plan for discharge could reduce service users' motivation to develop coping skills, affect the service's capacity to take on new referrals, and encourage 'dependency'. These clinicians felt having discharge or self-sufficiency as a time-specific goal from the beginning of care was helpful. However, other clinicians in the same study favoured offering continuing support at a lower level of intensity (for example through peer support), rather than absolute discharge following a period of intensive treatment, with clear provisions for re-engaging with services if required.

Short-term therapy, such as that offered by IAPT in the UK, tended to be seen as insufficiently flexible and intensive for people with CEN. In one study, primary care clinicians described a sense that they were "short-changing" service users [64]. In a few studies, clinicians also expressed fears that short-term support could potentially be harmful or experienced by service users as 'abandoning' and 'retraumatising'. However, in a small number of studies clinicians did argue that short-term support had value, either at specific points in service users' treatment journeys, or for those with less severe difficulties.

Clinicians in multiple studies also underlined the need to deliver both psychotherapeutic interventions and pragmatic social support to meet the varied and fluctuating needs of this population. Pragmatic support, which was reportedly offered more often in specialist services, could include vocational, educational, social, substance misuse, or parenting support, as well as skills to promote independence.

**Intervention models.** Specific treatment models that clinicians reported as having therapeutic benefits included Dialectic Behaviour Therapy (DBT) [51,55,60,61], Mentalisation

Based Therapy (MBT) [44,55], Cognitive Analytic Therapy (CAT) [68] and psychodynamic formulations [58]. However, in several studies, clinicians also emphasised that 'one size does not fit all', that diverse, flexible treatment options were needed within mental health services and in primary care, and that more formulation-driven treatments could be more beneficial than those based on diagnosis or driven by manuals.

There was a consensus across studies that a variety of approaches could be taken to some core therapeutic tasks, making a range of interventions similarly effective in achieving good outcomes. Clinicians tended to see difficulties with managing emotions as central in CEN, and prioritised interventions that promoted development of skills relating to emotion regulation, distress tolerance, or developing a capacity for *thinking* and *feeling* rather than *doing*. Similarly, models that helped service users to practice their interpersonal skills (e.g., via groups, peer support, or therapeutic communities) were seen as valuable in several studies. DBT was the specific therapeutic intervention most often discussed in studies and clinicians identified several benefits from this. As well as helping service users develop better relationships and emotion regulation, clinicians felt it was based on a clear model and manual, and that it promoted hope, decreased medication use, encouraged service users to take responsibility for treatment, and helped encourage compassion, understanding and team working on the part of clinicians. Clinicians in some studies did, however, also report that delivering DBT placed considerable demands on them and their services, including the need for intensive training, implementation of a complex model allowing relatively little flexibility, and being contactable outside of working hours.

Formats like groups, peer support, and therapeutic communities were also valued for broadening the range of available options and promoting collaborative, user-led models of care and empowering service users to have ownership over their treatment in a more democratic way. Finally, support for family and friends was identified in several studies as important, but as an area where even well-resourced specialist services often fall short despite the perception that people with CEN often experience difficulties with relationships.

## 4. Managing safety issues and crises: Being measured and proactive

Managing safety issues was considered vital across all treatment settings. The nature of deliberate self-harm and other safety issues in the context of CEN was seen as differing from acute presentations in other mental health conditions because of its chronic, recurrent and to some extent predictable nature. As such, clinicians felt it could be prepared for proactively, through open dialogue with service users to agree parameters within which clinicians would respond.

In a small number of studies, clinicians suggested that 'rescuing' or stepping in too quickly at times of crisis could be detrimental or disempowering for service users. However, there was a competing need not to become neglectful, with a lack of consensus regarding how available clinicians should make themselves. Views about out of hours service provision varied. In one study of community-based mental health services implementing DBT, some clinicians described 24/7 availability or an 'on call' system as a 'step backwards' and ineffective. But in other studies, clinicians argued that this was important, and that greater availability of support in fact usually reduced the need for it. Some clinicians felt that people with CEN were seen as 'bad' for posing a safety risk, in contrast to those with other diagnoses, such as psychosis, who were seen as 'mad'.

Practice in mainstream services was described in some studies as risk-averse and reactive, sometimes creating a vicious cycle wherein service users felt they had to present in crisis to get more input. Clinicians used to dealing with crises in the context of conditions such as depression or psychosis were reported to struggle to manage the specific dynamics of safety concerns

for people with CEN. Specialist services were seen as adopting more proactive approaches, negotiating plans for managing safety issues in collaboration with service users, moving away from action-reaction or fearful responses from clinicians, and fostering ownership of the management of safety issues among service users.

## 5. Clinician and wider service needs: Whose needs are they anyway?

**Clinician needs.** A recurring challenge across studies was for clinicians to reconcile their own needs with those of service users. This dilemma was particularly acute where clinicians lacked organisational support or adequate supervision. Clinicians found themselves negotiating between meeting the needs of service users, their own needs, and wider service needs. When synthesising studies, it was complex at times to disentangle whose needs were in reality met by particular practices. For example, when clinicians described a need to reduce service users' alleged 'dependency' and promote 'self-sufficiency', this seemed in part connected to clinicians' own feelings of being overwhelmed, as well as to wider service pressures to conserve resources. One study of clinicians working in "personality disorder" services [38] suggested that service users' perceived difficulties (e.g., with reflection) could be 'mirrored' further up the organisation. This study also reported that service leads across several teams appeared to be 'charismatic' but also 'autocratic', seeking to 'quell dissent' among clinicians by adopting firm, unequivocal stances. In other studies, it was clear that services, rather than service users, were at times experienced by clinicians as 'difficult to engage with'.

> *It's not the patients that make you frustrated nowadays, it's the organization around that is troublesome.*

> DBT Therapist (Perseius et al., 2003) [61]

The importance of clinicians feeling supported in their work was a common theme across studies. Working effectively with people with CEN without becoming burnt out was seen as achievable, but the organisational support needed to do so was often missing, with the low priority and investment accorded to treatment of people with CEN affecting both service users and clinicians. Clinicians valued both supportive relationships with colleagues and formal supervision in a variety of formats, including individual and whole team supervision and input from external experts. The importance of addressing clinicians' own emotional needs, engaging in reflective practice and enabling clinicians to process their own vulnerabilities and 'destructive' emotions was emphasised, but provision was frequently described as inadequate.

Good team-working and sharing responsibilities for treatment and decisions regarding safety also helped clinicians to feel supported. This appeared to be reported most often regarding specialist teams, especially in those using DBT and CAT models, and in therapeutic communities, and least frequently in primary care settings–where "you're kind of left on your own with somebody" [64]. There could also be challenges in teams where only one or two clinicians in a team were trained in a particular therapeutic intervention or skill set. While clinicians saw value in including a range of clinicians with diverse backgrounds and approaches, they also felt this could encourage "splitting", making it more difficult to develop a shared language or model of understanding across team members.

Having divided caseloads (i.e., not fully CEN) was considered by some to be beneficial for integration of CEN work into generic teams and for staff wellbeing. However, having competing clinical priorities could impede therapeutic work, and the 'psychological shift' between various roles was experienced by some as challenging. One study noted that specialist services tended to promote broad, combined roles where all clinicians contributed to delivering the

therapeutic model, but this required significant training. Specialist services sometimes had 'flat hierarchies' which could be empowering but also frustrating for clinicians when responsibility was equal but authority or pay, for example, was not.

**Interagency working and the wider system.** Effective inter-team and inter-agency working was considered important for management of the resource-intensive, multi-agency, and often out-of-hours service use by people with CEN. However, reports of inadequate communication between services were common at all levels of care. Challenges included high staff turnover, staff cutbacks due to reduced budgets, time constraints, and disagreements between clinicians or competing priorities, with poor interagency working leaving clinicians feeling more anxious and less contained. Pre-existing, personal, or good professional relationships [63] and clearly assigned responsibilities [47] (taking into account service user preferences regarding clinicians and services where possible) facilitated interagency working.

Clinicians in mainstream services reported in several studies that they valued support from specialist services, such as in hub and spoke models, where specialist staff provide expert assessments, case consultation, supervision, and staff training to mainstream services [47,57,58,71]. This model was perceived as making efficient use of specialist staff, allowing them to support not only those on their small caseload for intensive therapy but also a much wider group beyond the dedicated services. However, reservations about such models were described in a few studies, including that specialist input from specialists could undermine professional roles in mainstream service, may be ineffective on an ad hoc rather than sustained basis, and risks specialist clinicians having unsustainable workloads. There were also some tensions identified between mainstream and specialist services, where mainstream services were seen as having to 'firefight', whereas specialist services were perceived to have greater freedom to 'select' service users, refuse certain responsibilities, and prioritise time for reflection.

**Establishing new services, interventions and skills.** Finally, a number of studies were conducted in the context of establishing a new service or intervention programme, and thus themes emerged relating to good practice in initial implementation. Factors that were considered helpful for developing new services or interventions included: managerial support, recruitment of appropriate staff, leadership that embraced uncertainty and allowed clinicians freedom to innovate, team building, cross-agency and whole team training, and having realistic plans, timescales and budgets. Ongoing sustainability of new services was facilitated by integrating them into existing service systems, effective interagency working, and measuring and demonstrating good outcomes. Clinicians trained in new models described feeling like 'beginners' despite their clinical expertise and being required to make significant time commitments for implementation and ongoing practice and learning. There was widespread recognition of the need for ongoing support and training beyond the initial phase to support knowledge retainment and ensure programme sustainability. Some questioned the suitability of mental health service settings for delivering services given previous unsatisfactory or traumatic experiences for service users. However, acquiring alternative premises was often challenging.

## Discussion

Several overall proposals can be drawn from this synthesis of clinicians' perspectives for good practice in treating people with CEN effectively and respectfully and at the same time supporting the clinicians working with them. Areas of consensus between the findings of eligible studies included the need for high quality, holistic assessments and care plans encompassing physical, psychological and social needs; easily navigable referral systems enabling good continuity of care; and the need for a proactive, collaborative approach to safety management. Therapeutic relationships were seen as key and as a major common factor in the success of

different approaches, and clinicians in participating studies believed that they could be improved through greater therapeutic optimism, overcoming pejorative attitudes, developing partnerships between service users and clinicians through shared responsibility and decision making, radical acceptance and a non-expert stance, and sustainable models for service user involvement in care.

Some dilemmas and variations in opinion were also identified, especially regarding the balance between doing 'too much' or 'too little.' Potential positive and negative consequences were identified both for open-ended long-term input and for time-limited input, as for 24-hour availability of clinicians in specialist services. Those who advocate for long-term support may be more in tune with service users, reported often to see periods of treatment as too short and continuing support between periods of intensive therapy as lacking [17]. Whether or not services were time-limited, there was agreement that careful collaborative discharge planning was required to mitigate some of the frequently experienced challenges and help service users work towards self-sufficiency.

Many of these findings align with those identified in our accompanying meta-synthesis of the perspectives and experiences of service users with CEN [17]. For example, service users also appear to prioritise individualised care, preferring clinicians to focus on individual needs and aspirations rather than diagnosis or intervention fidelity. Clinicians were called upon in papers on service user perspectives to sustain hope and provide encouragement while at the same time maintaining realistic expectations and not invalidating service user distress. The centrality of the therapeutic relationship is a further point of consensus. While both clinicians and service users emphasised the need to offer a variety of treatment options to meet service users' heterogeneous needs, service users also prioritised structure, stability and a long-term perspective in their care. These are not inconsistent demands as options can be flexible and varied, yet their delivery can remain structured and consistent on an individual level. Whilst discontinuity of care and difficulties accessing services are often reported elsewhere for other diagnostic groups as well, we suggest that clinician reports in our synthesis reinforce views from service users [17] and policy makers [23] that this group is especially poorly served in terms of a service system designed to be accessible and meet a range of needs.

Concerns around the usefulness and impact of using "personality disorder" labels were also similar to those reported from studies of service user perspectives. However, the included papers on clinician perspectives tended less to reflect recent calls by service user advocates and some clinicians, supported by patient testimonials and growing evidence, to give trauma a central role in the assessment and treatment of CEN, a call also reinforced by feminist critiques of "personality disorder" as a mechanism for pathologising natural responses to oppression, abuse and structural inequalities [18]. This omission may in part reflect the fact that most studies were conducted before the rise of the 'Trauma not PD' movement [72,73]. We suggest that alongside the priorities identified above, incorporating trauma-informed approaches to care and preventing re-traumatisation within mental health settings should be seen as key elements in good practice if a shared agenda for service improvement is to be agreed on by service users and clinicians [74].

Exploring clinician perspectives is particularly valuable for identifying ways of promoting positive change and for removing clinician-related barriers to this. This review echoes much other literature in identifying pejorative clinician attitudes and behaviours as an important obstacle to delivering care that is even adequate, especially in non-specialist settings. Developing and evaluating ways to challenge and change such behaviours is thus a pressing need. This review also identifies the need to extend more support to clinicians working with people with CEN; across several studies, clinicians reported on the significant emotional toll of their work, which could potentially fuel negative behaviours and a lack of therapeutic optimism. Several of

our themes related to the need for clinicians to strike a balance, including balancing connection against distance, doing too much or too little in terms of treatment provision and balancing service user empowerment and independence with service pressures of risk-aversion. Needs of different stakeholders also require balancing: for example, do some clinicians warn against long-term input for the benefit of service users (to promote independence), for the benefit of themselves (to avoid challenging work), or for the benefit of services (to meet capacity constraints)? This balancing act, together with caseload and referral pressures, may well contribute to the emotional toll of working with people with CEN. However, clinicians, especially in specialist services, also described many ways of alleviating this burden, including through supervision, reflective practice and informal support between colleagues. The burdens associated with difficult therapeutic decisions, especially regarding safety, were clearly alleviated by being shared, both with colleagues and service users. As such, multidisciplinary co-produced formulations, maintaining the centrality of the therapeutic relationship, and 'holding in mind' the service user could provide some guiding principles for clinicians when navigating these complex balances and would be a useful focus for further research.

Constraints on good practice relating to the wider service system were recurrently described, including exclusive thresholds and referral pathways, inflexibility of services to meet diverse and long-term needs and manage co-occurring conditions, and lack of time for reflection and training. Lack of recognition of the needs of people with CEN and lack of resourcing to meet these needs were widely reported and likely to contribute. These deficits may also reflect a lack of evidence and strategic thinking on how to optimise service design to result in coherent pathways allowing smooth transitions between accessible services corresponding to service users' needs and delivery of a full range of evidence-based psychosocial interventions in all relevant settings. This will require design of the system so that relevant evidence-based interventions can be delivered in primary care, generic secondary and specialised services, with smooth transitions and collaborative working between all sectors, including support for primary care from specialised CEN services. The major focus of research on CEN has been on the effectiveness and cost-effectiveness of relatively short-term psychological therapies: co-produced research taking a whole-system perspective on how to design systems of care that meet the varying needs of diverse service users at different stages in their pathways through services now appears to be an important need.

## Limitations

We aimed to include papers regarding management in the community of people with a range of "personality disorder" diagnoses or who might have related difficulties, such as recurrent self-harm, but not have received such a diagnosis. However, in practice most studies focused on people who had received a diagnosis of "borderline personality disorder". As such, our findings relate mainly to this group, with some heterogeneity in the ways in which study samples were identified. Our search criteria were broad, encompassing qualitative literature using all methods on all aspects of community care for all personality diagnoses: we therefore made a pragmatic decision to exclude papers that had not been peer-reviewed, were not in English, and dissertations or theses. This may have resulted in substantial contributions being missed. There was a good variety of professional backgrounds and levels of care across included papers, but little literature about voluntary organisations and other community services outside the secondary mental health care system. This may reflect limitations of the search strategy, but probably also indicates a scarcity of research in these areas. This may mean that the voices of staff who support individuals who have disengaged or been excluded from the mainstream mental health system are not included.

As this is a meta-synthesis identifying and cross-validating over-arching themes across many studies, a level of nuance and specificity will have inevitably been lost, with findings pooled from a variety of contexts, dates and countries. The two researchers who worked the most closely on synthesis (JT and BLT) both have clinical experience of providing mental health care, while three other authors (JR, TJ, EB) bring relevant lived experience of service use–the results presented here and their interpretation may well be shaped by their perceptions born from these experiences. Efforts were made to counter this through adopting an inductive approach to analysis, double coding a portion of papers, discussing themes together and iteratively, and through the collaboration of the review team and experts by experience and occupation.

## Conclusion

Clinicians' experiences of and perspectives on good practice for providing community care for people with CEN offer valuable insights into how to better meet the needs of this population and the needs of the clinicians supporting them and are largely in harmony with the perspectives of service users [17]. In further research, a focus is now needed on how to implement these principles of good practice across the service system to improve service user outcomes and the experiences of service users and clinicians. Previous research has tended to focus on individual psychological interventions: a focus on designing a whole system of care that can meet the longer-term needs of people with CEN in a sustainable way is now desirable. Development and evaluation of fidelity measures that reflect agreed good practice [75,76], and of approaches to support services in achieving and maintaining high fidelity, is a potential approach to meeting this need. The apparent congruence on many values and principles between service users and clinicians suggests that a co-produced approach to future research, service development and policy formulation is likely to be fruitful. Finally, an overarching emerging issue deserving further research and policy development is of equity: clinicians echo service users in arguing that people with CEN tend to be a marginal group, often not prioritised for resources and attracting negative attitudes and behaviour. Change is not likely to be achieved unless the needs of people with CEN are placed on an equal footing with the needs of people with other long-term physical and mental health conditions.

## Lived experience commentaries

In line with service user critiques and our own lived experience, this meta-synthesis provides further evidence that for many people with CEN, current mental health services are simply not fit for purpose. From clinician burnout and pejorative attitudes, to a clinical victim-blaming culture when a service cannot meet service users' needs, the signs of a system at breaking point are undeniable.

Since clinicians themselves seem to recognise the wider social context, i.e., that trauma and adversity are major contributors to the distress experienced by people with CEN, it begs the question: why do most services still regard the medical model as the panacea? It appears that we need major systemic change and services should truly embrace inclusive, co-designed approaches that value lived experience and also support user-led models of care.

Clinicians' concerns around diagnostic utility are noted and shared. However, 'dancing around the diagnosis' due to fears of stigma and exclusion—no matter how well intentioned—may actually be counterproductive and inadvertently further perpetuate the stigma. It only underscores the urgent need to address this controversial terminology.

Despite the awareness of a gender bias that results in women with CEN being disproportionately more likely to receive a "borderline personality disorder" label than men, there is no

mention of the overlap with Autism Spectrum Conditions (ASC) [77] and the fact that women are conversely under-diagnosed with ASC [78]. This can have serious implications for potentially mis-diagnosed service users who may end up trapped on unsuitable treatment pathways and therefore constitutes a significant gap in the evidence base warranting investment in further research.

While we support inter- and multi-agency working in principle, stakeholders need to be mindful of its potential pitfalls. For example, as if pathologising legitimate feelings of distress wasn't problematic enough, collaborating with law enforcement (e.g., through the "Serenity Integrated Mentoring" programme, a widely criticised intervention implemented in England, which integrates police officers in community mental health teams and routinely denies so-called 'High Intensity Users' access to crisis care [79,80]) can exacerbate the risk of going as far as criminalising CEN [81]. Such misconceived interventions can not only permanently destroy service users' trust in mental health services, but can also have absolutely devastating effects on their life chances, negating any attempt at meaningful recovery.

Overall, it is encouraging that there *are* clinicians who share our views after all, and the answer to "Whose needs are they anyway?" should be a resounding "Everyone's!"

After all, service users don't benefit from working with stressed and burnt-out clinicians, either; therefore, the desire to improve staff training and support is mutual. Unfortunately, the prevailing systemic flaws are not conducive to either individual practitioner or service improvement. Likewise, influencing those clinicians who are steadfast in holding onto stigmatising views of people with CEN is going to be a major challenge that must be addressed with co-production throughout service development and delivery.

Eva Broeckelmann and Jessica Russell

## Broken Mirrors

Whilst reading this review, I was struck by the allegory of a mirror. The focus is on clinicians, but its sister paper with a service user focus [17] reflects the same issues. The mirror allegory goes beyond similar themes being reflected. The opinions of each side are fragmented–like a broken mirror. The broken fragments of each side appear as perfect replicas of the other, yet can only see each other in reverse, appearing as polar opposites.

The data here is constricted to what is within the literature, with both papers dutifully reporting this. This data is limited in providing an understanding of why, despite appearing to want the same thing, there is such a relational divide between service user and service provider.

The roles of people working within the Lived Experience Professions (i.e., peer support workers, service user consultants, lived experience researchers) could be described as roles that bridge between the two polarised worlds, communicating sameness and difference between the two. Literature exploring how this could relate to developing relational bridges within the field of trauma/complex emotional needs/"personality disorder" is not included–potentially because it does not exist or exists in a format that does not fit within the search criteria. This highlights the importance of being able to value experiential data as a valid consideration within research, in order to lessen the phenomenon of studies giving a perfect view of one small fragment of the broken mirror, whilst disregarding the rest. Services benefit more from a full view of the broken mirror, even if the individual shards are more blurred than one perfect piece.

This gave me pause for thought when researchers described their experiences of working in services as a potential limitation in the review. Once they have acknowledged their own

perspective, understanding the line between this and the data, their 'limitation' is in fact a strength–and this knowledge needs to be recognised, valued and encouraged more. The literature we use to inform and shape policy is not being practiced under lab conditions, but in the messy world where broken mirrors exist.

Tamar Jeynes

## Supporting information

**S1 Checklist.**
(DOC)

**S1 Table. Quality appraisal.**
(DOCX)

**S2 Table. Supporting quotes.**
(DOCX)

**S1 Appendix. Search strategy.**
(DOCX)

**S2 Appendix. Eligibility criteria.**
(DOCX)

## Acknowledgments

This report is based on independent research commissioned and funded by the National Institute for Health Research Policy Research Programme. The views expressed are those of the author(s) and not necessarily those of the NHS, the National Institute for Health Research, the Department of Health and Social Care or its arm's length bodies, and other Government Departments.

## Author Contributions

**Conceptualization:** Sian Oram, Sonia Johnson.

**Data curation:** Jordan Troup, Thomas Steare, Zainab Dedat.

**Formal analysis:** Jordan Troup, Billie Lever Taylor.

**Methodology:** Jordan Troup, Billie Lever Taylor.

**Supervision:** Billie Lever Taylor, Luke Sheridan Rains, Chris Cooper, Sian Oram, Sonia Johnson.

**Writing – original draft:** Jordan Troup, Billie Lever Taylor, Eva Broeckelmann, Jessica Russell, Tamar Jeynes.

**Writing – review & editing:** Billie Lever Taylor, Luke Sheridan Rains, Eva Broeckelmann, Jessica Russell, Tamar Jeynes, Chris Cooper, Shirley McNicholas, Sian Oram, Oliver Dale, Sonia Johnson.

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
