## [Decision Letter · Decision Letter 0]

16 Jun 2021

PONE-D-20-39422

Clinician perspectives on what constitutes good practice in community services for people with Complex Emotional Needs: A qualitative thematic meta-synthesis

PLOS ONE

Dear Dr. Sheridan Rains,

Thank you for submitting your manuscript to PLOS ONE. After careful consideration, we feel that it has merit but does not fully meet PLOS ONE’s publication criteria as it currently stands. Therefore, we invite you to submit a revised version of the manuscript that addresses the points raised during the review process.

The manuscript has been evaluated by two reviewers, and their comments are available below.

The reviewers have raised a number of concerns that need attention. They request additional information on methodological aspects of the study , and revisions to the Introduction and Discussion sections. 

We look forward to receiving your revised manuscript.

Kind regards,

Carmen Melatti

Staff Editor

PLOS ONE

Journal Requirements:

Additional Editor Comments (if provided):

Reviewers' comments:

Reviewer's Responses to Questions

**Comments to the Author**

1. Is the manuscript technically sound, and do the data support the conclusions?

Reviewer #1: Yes

Reviewer #2: Yes

2. Has the statistical analysis been performed appropriately and rigorously? 

Reviewer #1: N/A

Reviewer #2: N/A

3. Have the authors made all data underlying the findings in their manuscript fully available?

Reviewer #1: Yes

Reviewer #2: Yes

4. Is the manuscript presented in an intelligible fashion and written in standard English?

Reviewer #1: Yes

Reviewer #2: Yes

5. Review Comments to the Author

Reviewer #1: This manuscript synthesizes qualitative evidence on good practice in community mental health services for people with Complex Emotional Needs (CEN).

It is a well written manuscript, with a clear focus that uses a sound methodology, and which reports clearly its results. It brings to light important issues regarding mental health services for people with CEN. The 6 over-arching themes are well presented, and the discussion brings interesting points. It is about the perspectives of a vast variety of clinicians and contrasts these opinions with services users’ perspectives by reporting some results of another manuscript by the same research group, as well as including lived experience commentaries.

I would suggest only minor modifications.

It states from the beginning that the synthesis is about ‘community mental health services’. They should be defined as this expression can mean somewhat different types of services throughout the world. It appears here to include generic and specialist settings whereas elsewhere it might only include non-specialist settings. ‘Generic services’ should also be defined.

In the eligibility criteria (methodology), thesis and dissertations have been excluded from the systematic literature search, but it is not clear why.

It is intriguing that despite no limit in the search being put on language, all the studies included are from English speaking countries, except for Sweden. This appears to be a limit to the review.

In the discussion, it could have been developed how primary care settings and specialized mental health settings could further collaborate in providing care for this population of services users. With the prevalence of CEN, it is not realist to think specialized service can provide all the required care and adapted models of collaborative mental health care are needed for CEN services users, different from the stepped care approach.

It would also be interesting to know how services using trauma-informed models have resolved the challenges regarding providing continuous enough care while responding to the population needs.

In the lived experience commentaries, the reference to ASC would deserve further development. Also, for unfamiliar readers from outside England, it would help to have a brief description of the ‘Serenity Integrated Mentoring Program’, although the general idea is well understood.

Reviewer #2: Thank you for asking me to review this manuscript. My comments are below:

1. There are a huge number of authors for this paper. It would be helpful to itemize what author has done what (found it thanks �).

2. Please clearly cite the basis for the term CEN. This is critical. Please also make explicit that your included persons are those with PD and how this diagnosis is made (there is a very wide variation in the literature). If the reader is unclear about the population included they will not be able to follow the paper.

3. After examining references 27-32 I think you need to drop CEN and replace this with PD throughout the paper, or write an accompanying editorial to justify and critique the term CEN. (I note for example that cognitive and behavior issues are central, as are interpersonal ones in the diagnosis of PD and your term doesn’t seem to capture this).

4. Justify the 2003 starting date

5. Cite endnote

6. Define ‘good’ and ‘cen’ in your eligibility criteria.

7. Was any triangulation carried out? You seem to have tons of authors but in effect only one author has done most of the coding. This is a significant weakness that needs to be acknowledged.

8. Described your synthetic approach. Just citing 41 is insufficient- what process did you follow? (Lets pretend I want to replicate your work, how can I do that on the basis of this paper)

9. Make the quality assessment (table 2) a supplemental table. Its distracting

10. Please add more data to the opening para of the results. I want to know what you’ve got from where far more accurately.

11. Make table 3 a supplement. I do like the use of multiple quotes from multiple papers to justify the analysis but do not think it adds anything to the paper itself.

12. The discussion of theme one seems to me to be about stigma related to the diagnosis, as opposed to its use and misuse. Please consider revising the title. (Also you can’t relabel in the intro and then have this as your first finding- it’s the tail wagging the dog).

13. I suspect theme 2 could be the case for every mental health, and possibly any medical sub-specialty. You will need to address this in the discussion (the wider medical and psychiatric content).

14. Theme four is surely ‘dialectics’, not doing too much or too little…..

15. lines 360 to 362 need citations

16. Theme 5 is fantastically presented, thank you

17. Please cite lines 464 to 469

18. I find totally novel ‘data’ in the discussion a bit distracting. For examples the bits about service user views and the trauma and feminist comments are a bit out of kilter. They could be reframes to provide a sociological (and possibly) biological construct to understand the data somewhat better.

19. I like the service user reflections.

6. PLOS authors have the option to publish the peer review history of their article (what does this mean?). If published, this will include your full peer review and any attached files.

Reviewer #1: No

Reviewer #2: **Yes: **Giles Newton-Howes

---

## [Author Response · Author response to Decision Letter 0]

9 Jan 2022

Dear PLOSONE

Re: [PONE-D-20-39422R1] 

Thank you for the invitation to revise and resubmit our manuscript “Clinician perspectives on what constitutes good practice in community services for people with Complex Emotional Needs: A qualitative thematic meta-synthesis”, in response to the valuable editorial and reviewer comments. We have detailed our responses to each point, and the changes made to the manuscript, below.

We hope that the manuscript is now suitable for publication in PLOS ONE and look forward to hearing from you further.

Yours sincerely,

Luke Sheridan Rains, Sonia Johnson, Jordan Troup, Billie Lever Taylor, Eva Broeckelmann, Jessica Russell, Tamar Jeynes, Chris Cooper, Thomas Steare, Zainab Dedat, Shirley McNicholas, Sian Oram, and Oliver Dale.

Editorial comments from 19/11/2021

1. We note that several of your files are duplicated on your submission. Please remove any unnecessary or old files from your revision, and make sure that only those relevant to the current version of the manuscript are included.

Response – we have now removed the duplicate files

2. Thank you for stating "Funding" section of your manuscript:

"This paper presents independent research commissioned and funded by the National Institute for Health Research (NIHR) Policy Research Programme, conducted by the NIHR Policy Research Unit (PRU) in Mental Health (grant no. PR-PRU-0916-22003). The views expressed are those of the authors and not necessarily those of the NIHR, the Department of Health and Social Care or its arm’s 591 length bodies, or other government departments. The funders had no role in study design, data collection and analysis, decision to publish, or preparation of the manuscript."

"This paper presents independent research commissioned and funded by the National Institute for Health Research (NIHR) Policy Research Programme, conducted by the NIHR Policy Research Unit (PRU) in Mental Health (grant no. PR-PRU-0916-22003). The views expressed are those of the authors and not necessarily those of the NIHR, the Department of Health and Social Care or its arm’s 591 length bodies, or other government departments. The funders had no role in study design, data collection and analysis, decision to publish, or preparation of the manuscript."

Response - We have now removed the funding information from the manuscript and included it in our cover letter as requested. This statement is tracked in the cover letter to make it clear how the letter has changed since our original submission. The funding statement that you provided is correct and to eliminate ambiguity is repeated below:

"This paper presents independent research commissioned and funded by the National Institute for Health Research (NIHR) Policy Research Programme, conducted by the NIHR Policy Research Unit (PRU) in Mental Health (grant no. PR-PRU-0916-22003). The views expressed are those of the authors and not necessarily those of the NIHR, the Department of Health and Social Care or its arm’s 591 length bodies, or other government departments. The funders had no role in study design, data collection and analysis, decision to publish, or preparation of the manuscript."

---

## [Editor Report · Decision Letter 1]

18 Apr 2022

Clinician perspectives on what constitutes good practice in community services for people with Complex Emotional Needs: A qualitative thematic meta-synthesis

PONE-D-20-39422R1

Dear Dr. Sheridan Rains,

We’re pleased to inform you that your manuscript has been judged scientifically suitable for publication and will be formally accepted for publication once it meets all outstanding technical requirements.

Kind regards,

Marianna Mazza

Academic Editor

PLOS ONE